# The Effect of 2′F-RNA on I-Motif Structure and Stability

**DOI:** 10.3390/molecules30173561

**Published:** 2025-08-30

**Authors:** Cristina Ugedo, Arnau Domínguez, Irene Gómez-Pinto, Ramon Eritja, Carlos González, Anna Aviñó

**Affiliations:** 1Instituto de Química Física Blas Cabrera (IQF), CSIC, Serrano 119, 28006 Madrid, Spain; cugedo@iqf.csic.es (C.U.); irene_gomez@iqf.csic.es (I.G.-P.); 2Instituto de Química Avanzada de Cataluña (IQAC), CSIC, Jordi Girona 18-26, 08034 Barcelona, Spain; adztnt@cid.csic.es; 3Centro de Investigación Biomédica en Red de Bioingeniería, Biomateriales y Nanomedicina (CIBER-BBN), 28029 Madrid, Spain

**Keywords:** i-motif, NMR, non-canonical nucleic acids, XNA, fluorine substitution

## Abstract

I-motifs are non-canonical, cytosine-rich DNA structures stabilized by hemiprotonated C•C^+^ base pairs, whose formation is highly pH-dependent. While certain chemical modifications can enhance i-motif stability, modifications at the sugar moiety often disrupt essential inter-strand contacts. In this study, we examine the structural and thermodynamic impact of incorporating 2′-fluoro-ribocytidine (2′F-riboC) into i-motif-forming sequences derived from d(TCCCCC). Using a combination of UV, ^1^H NMR, and ^19^F NMR spectroscopy, we demonstrate that full substitution with 2′F-riboC strongly destabilizes i-motif, whereas partial substitutions (one or two substitutions per strand) support well-folded structures at acidic pH (pH 5). High-resolution NMR structures reveal well-defined i-motif architectures with conserved C•C^+^ pairing and characteristic interstrand NOEs. Sugar conformational analysis reveals a predominant North pucker for cytosines, which directs the fluorine substituent toward the minor groove of the i-motif. ^19^F NMR further confirms slow exchange between folded and unfolded species, enabling the simultaneous detection of both under identical experimental conditions and, consequently, highlighting the utility of fluorine at the 2′ sugar position as a spectroscopic probe. These findings provide insights into fluorine-mediated modulation of i-motif stability and further extend the utility of ^19^F NMR in nucleic acid research.

## 1. Introduction

I-motifs are four-stranded, non-canonical DNA structures usually formed by cytosine-rich sequences. They are composed of two parallel duplexes stabilized by hemiprotonated C•C^+^ base pairs. These duplexes intercalate in an antiparallel fashion, giving rise to the characteristic architecture of the i-motif [1]. Interest in i-motifs has grown considerably due to their potential roles in biological process and their promising applications in nanotechnology [2,3,4,5]. In biology, i-motifs are somehow associated to G-quadruplex, as they can form in sequences complementary to those capable of G-quadruplex formation. [6,7] This complementarity suggests a possible involvement in regulatory processes such as transcription or replication. [8,9,10] The recent observation of i-motif formation in vivo in regulatory regions [11], although still controversial [12], has sparked investigations into the i-motif as a potential target of pharmaceutical interest [5,13,14].

Beyond their biological relevance, i-motifs have also attracted attention in the field of nanotechnology [15], particularly due to their strong pH dependence [16,17,18,19]. The requirement for cytosine hemiprotonation (C•C^+^) for structure formation renders i-motifs highly sensitive to pH changes. The exact pH at which half of the structure is denatured (pH_1_/_2_) varies depending on the specific sequence and context [20,21,22]. Importantly, this pH responsiveness can be modulated through chemical modifications, opening the door to fine-tuning their stability and function [2].

Considerable effort has been devoted to understanding the impact of chemical modifications on i-motif stability. Modifying the sugar-phosphate backbone remains particularly challenging since inter-strand sugar-sugar contacts are critical to the stability of the i-motif [23]. Disruptions to these interactions typically result in destabilization [24,25,26]. Among the sugar modifications explored to modulate i-motif behavior, the position and stereochemistry of the 2′-substituent play a pivotal role. In the native i-motif, sugar-sugar contacts in the narrow groove, especially between back-to-back cytidines, are essential for structural stability. As such, modifications that alter sugar geometry or introduce steric hindrance often lead to destabilization. This is clearly seen with RNA [27,28] or 2′-O-methyl modifications, which place polar groups into the narrow groove, disrupting close sugar contacts. In contrast, locked nucleic acids (LNAs), which constrain the sugar to a C3′-endo pucker similar to RNA, can stabilize i-motifs, although high LNA content leads to conformational polymorphism [29].

Among the few modifications found to be stabilizing, 2′-fluoroarabinocytidine (2′-F-araC) stands out [30,31]. In the arabino configuration, 2′-substituents point toward the major groove, preserving the essential minor groove inter-strand contacts. Modifications like araC [32] or 2′-Me-araC [33] are generally well tolerated, but 2′-F-araC provides a unique stabilizing effect. This is likely due to the high electronegativity of fluorine, which alters the sugar’s charge distribution and can facilitate additional electrostatic and hydrogen bonding interactions [30,34]. These studies highlight that the effect on stability is not merely a function of the sugar’s pucker, but also of how the 2′-substituent projects into the grooves.

In this study, we investigate the effect of fluorine substitution in the opposite stereochemical configuration: the ribo configuration (Figure 1a). In contrast to DNA, the presence of riboses in the sequence has been shown to destabilize i-motifs, likely due to the unfavorable positioning of the 2′-OH group into the i-motif’s minor groove [27]. By extension, 2′-F-RNA could also be destabilizing, although the smaller fluorine atom may exert a less severe effect than the hydroxyl group. Intriguingly, earlier studies by Fenna et al. [35], suggested otherwise, prompting a deeper structural investigation.

Here, we report the high-resolution three-dimensional structures of two i-motif variants derived from the sequence d(TCCCCC), each incorporating 2′F-riboC modifications. These structures provide insight into the molecular basis of the destabilizing effect induced by fluorine in the ribo configuration.

In addition to their impact on structure and stability, fluorine modifications are of significant interest due to their utility in ^19^F NMR spectroscopy. The incorporation of fluorine atoms as NMR-active probes offers a powerful tool for investigating nucleic acid structure and dynamics. [36,37,38] Fluorine probes, particularly at the C2′ position, have proven extremely useful for monitoring the formation of non-canonical nucleic acid structures. [39,40,41,42,43] Exploring 2′F-RNA modifications as ^19^F NMR probes in the context of i-motif structures further highlights the relevance of this study.

## 2. Results

### 2.1. I-Motif Formation and Stability in 2′F-Riboc Containing Sequences

Three sequences derived from d(TCCCCC), each containing C→frC mutations, were investigated (see Table 1). UV and NMR melting experiments were performed at various pH values (see Appendix A). No i-motif formation was detected at neutral pH in any of the sequences. At pH 6, weak imino signals between 15–16 ppm, corresponding to hemiprotonated C•C^+^ base pairs, are observed at T = 5 °C (see Appendix A). These signals are weak, and their number does not match that expected for a single well-defined i-motif structure. The presence of two thymine imino signals around 11.5 ppm confirms the coexistence of at least two species and is consistent with the UV melting profiles, which display subtle hyperchromic transitions at low temperatures (Appendix A). In contrast, at pH 5, the imino proton signals for FR1 and FR2 are sharper and more intense, indicating better-defined folded structures, whereas no such signals are detected for FRfull at this pH. These observations are in line with UV melting experiments: FR1 and FR2 show melting temperatures of 55.9 ± 0.1 °C and 64.9 ± 0.1 °C, respectively, while no detectable transition is observed for the FRfull sequence (Table 1, and Appendix A). We therefore conclude that the fully substituted sequence does not form an i-motif under pH conditions ranging from 5 to 7. Interestingly, i-motif formation for FRfull could be detected at pH 4, although the chemical shift dispersion is very poor (Appendix A). In contrast, FR1 and FR2 yield well-resolved spectra at pH 5, displaying sharp and well-dispersed signals consistent with a single well-folded i-motif structure (Figure 1c and Appendix A). ^19^F NMR spectra provided additional insights. Distinct signals from folded and unfolded species could be identified (−115 to −117 ppm, and −123 ppm, respectively), suggesting that the equilibrium between these conformations is slow on the ^19^ F NMR timescale in the whole range of temperatures in which folded and unfolded species coexist—a behavior consistent with previously studied fluorinated i-motifs [30,39]. Melting experiments, monitored by changes in ^19^F signal intensity across temperatures, revealed partial unfolding even at low temperatures (Figure 1d and Appendix A). As expected, the unfolded species became predominant at high temperatures.

### 2.2. NMR Assignments

Exchangeable proton regions in the 2D NOESY spectra are shown in Figure 2a and Appendix A. The number and pattern of hemiprotonated imino signals, along with cross-peaks to cytosine amino protons, are consistent with the formation of four C•C^+^ base pairs. Since only two amino–imino NOEs are observed for each imino signal, we conclude that the C•C^+^ base pairs are formed between equivalent cytosines. Additional NOEs characteristic of i-motif structures are evident, including H2′–amino cross-peaks and interstrand sugar–sugar NOEs. The latter are especially important for connecting cytosine spin systems through the minor groove. Sequential assignment could be initiated from the T1 spin system, which is connected to C6 via the methyl and H6 protons. Additional connectivities were traced along the sequence C6→C2→C5→C3 (Figure 3).

### 2.3. NMR Constraints and Structural Calculation

^19^F NMR spectra were particularly informative, yielding a significant number of ^19^F–^1^H HOE cross-peaks (see Figure 2b). A total of 274 and 300 distance constraints were obtained from NOESY and HOESY experiments for FR1 and FR2, respectively (see Appendix A for structural statistics).

In addition to NOE-derived restraints, a qualitative analysis of J-coupling constants from DQF-COSY spectra was performed. Except for the T1 residue, the J_1′–2′_ couplings were undetectable in most sugars, strongly suggesting a North (C3′-endo) sugar conformation. In contrast, the T1 residue showed clear J_1′–2′_ and J_1′–2″_ couplings, consistent with a South (C2′-endo) sugar pucker. Strong intra-residual H3′–H6 NOEs further confirmed the North conformation for the cytosine sugars.

The distance and torsion angle constraints were used in restrained molecular dynamics simulations to calculate the structures, as described in the Methods Section. The 10 resulting structures are shown in Figure 4, Appendix A. Both FR1 and FR2 structures are well defined, with an RMSD of approximately 0.5 Å. The final AMBER energies and NOE violations are low, with no distance violations exceeding 0.3 Å. The final coordinates have been deposited in the Protein Data Bank.

### 2.4. Structure Analysis

The high-resolution NMR structures of FR1 and FR2 show a remarkable similarity, with a root mean square deviation (RMSD) between the two ensembles of approximately 1.0 Å. Although this difference is small, it is still meaningful, as the internal RMSD within each ensemble is around 0.5 Å, indicating well-defined structures.

Consistent with the J-coupling data, all cytosine residues adopt a North (C3′-endo) sugar pucker, as indicated by low pseudorotation phase angles. Both FR1 and FR2 adopt a 3′E topology, wherein the 3′-terminal cytosines participate in the terminal C•C^+^ base pairs of the i-motif. In this structural arrangement, all inter-strand sugar–sugar contacts occur between deoxyribonucleotides or between 2′F-ribonucleotides and deoxyriboses; no direct contacts between 2′F-ribose sugars are observed.

The fluorine atoms introduced through the 2′F-riboC modifications are oriented towards the minor groove of the i-motif (Figure 5a). They are only partially buried within the groove and do not appear to form hydrogen bonds or favorable electrostatic interactions with positively charged groups. Although they may participate in direct solvent interactions, they are likely limited due to restricted solvent accessibility. The presence of highly electronegative fluorine atoms alters the local charge distribution of the sugar ring. While such effects may stabilize certain nucleic acid structures, like A-form duplexes [41], they appear to be destabilizing in this context, likely by disrupting favorable hydrophobic interactions between sugars though the minor grooves

## 3. Discussion

Our results demonstrate that full substitution with 2′F-riboC (iFRNA) prevents i-motif formation at neutral pH, confirming that this chemical modification is not stabilizing under physiological conditions. Nevertheless, partially substituted sequences (FR1 and FR2) still form well-folded i-motif structures at pH 5, and provide sufficiently high-quality NMR spectra for detailed structural analysis. This highlights a valuable application of 2′F-riboC residues: although not stabilizing, they serve as effective ^19^F NMR probes in folded i-motif contexts. Substitution with ribocytidine (iRNA) has been previously shown to destabilize i-motifs due to steric hindrance from the 2′-OH group positioned within the narrow groove [27]. Our structural data for iFRNA confirm this destabilizing trend, but with a less pronounced effect: FR1 and FR2 retain a defined 3′E topology and stable C•C^+^ base-pairing at low pH. In contrast, Snoussi et al. observed two topologies for r(UC5) [27], with the 5′E form being more populated, probably due to the formation of fewer unfavorable riboC-riboC contacts in the 5′E than in the 3′E topology. The presence of multiple conformers, lower stability, and broader spectra were attributed to steric hindrance from 2′-OH groups. At pH 5, our iFRNA structures are topologically uniform (3′E only), more homogeneous, and less prone to interconversion, reinforcing the idea that fluorine is less perturbing than hydroxyl when placed at the 2′ position. Although of not sufficient quality for full structural determination, NMR spectra of the fully substituted iFRNA sequence (FRfull) at pH 4.2 (Appendix A)—the same conditions used in the r(UC5) study—suggest the formation of a more stable i-motif than that observed for the RNA counterpart

In contrast to iFRNA, 2′F-araC-modified i-motifs (iFANA) are strongly stabilized, even at neutral pH. This modification promotes favorable sugar-sugar and electrostatic interactions across the narrow groove by positioning the fluorine atom into the major groove. In these structures, 2′F-araCs adopt a C2′-endo conformation which facilitates the formation of hydrogen bonds and electrostatic interactions with positively charged groups as citosine amino protons (see Figure 5). Consequently, iFANA structures show increased thermal and pH stability, rendering remarkably stable structures at pH 7. This makes 2′F-araC unique among sugar-modification in i-motifs. Thus, the behavior of iFRNA can be positioned between iRNA and iFANA. It is less disruptive than ribose (2′-OH), allowing structural definition at low pH, but lacks the stabilizing features of the arabino configuration.

Fluorine atoms in our iFRNA structures are oriented toward the minor groove and do not appear to participate in direct hydrogen bonding. Although the fluorine atoms fit reasonably well within the groove and no significant steric clashes are observed, their close proximity to the phosphate groups in such a confined space makes electrostatic interactions unfavorable (see Figure 6). Nonetheless, favorable interactions with solvent cations at higher ionic strengths cannot be entirely ruled out. Our structural data suggest that the primary effect of fluorine substitution is to disturb the electronic distribution of the sugar ring, thereby weakening the hydrophobic sugar–sugar interactions that are essential for i-motif stability. This behavior contrasts with that of iFANA, where the fluorine atom resides in the major groove and can contribute to stabilizing electrostatic and hydrogen bonding interactions [30].

The role of electrostatic interactions in densely packed regions of nucleic acid molecules, such as those occurring between sugars in i-motifs, has long been discussed [23,44], and it is not surprising that apparently subtle differences, such as those between FANA and FRNA, can provoke opposite effects. Moreover, it has also been proposed that the stabilization of FRNA relative to its RNA counterparts may arise from changes in hydration and nucleobase polarization induced by fluorine, which in turn could enhance base-pair strength [45,46]. Although these effects have been described in the context of double-helical RNA structures, their applicability to i-motifs cannot be ruled out.

Beyond the structural implications 2′F-substitutions in i-motif structures, this study is an excellent example on how ^19^F NMR develops as a highly sensitive tool for detecting nucleic acid conformations. The high sensitivity of fluorine, combined with the absence of endogenous fluorine in biomolecules, provides excellent signal-to-noise ratio and minimal background. In our study, ^19^F NMR enabled the simultaneous observation of folded and unfolded i-motif species, revealing slow exchange dynamics that would be challenging to capture using traditional proton-detected techniques. This approach is particularly powerful for i-motifs [47,48] and other non-canonical structures [49] that often exist in equilibria with other conformations. This may be particularly relevant for live-cell imaging applications that allow real-time observations of cellular processes and molecular interactions [50,51,52,53]. Previous work on fluorinated nucleic acids, such as 2′FANA and 2′F-RNA in duplexes and G-quadruplexes, has also shown the advantage of using ^19^F NMR in probing conformational preferences and transitions, as well as characterizing partially folded species in complex equilibria. Thus, 2′F substitutions not only provide structural insight but also open valuable spectroscopic windows for monitoring nucleic acid behavior in solution.

## 4. Materials and Methods

### 4.1. Oligonucleotide Synthesis and Purification

5′-Dimethoxytrityl-N-acetyl-2′-deoxy-2′-fluorocytidine-3′-[(2-cyanoethyl)-(N,N-diisopropyl)]-phosphoramidite) were obtained from commercial sources (Glen Research, Sterling, VA, USA). Oligonucleotides were synthesized at 1 μmol scale with an H-8 DNA/RNA synthesizer (K&A Laboratories, Schaafheim, Germany). After the solid-phase synthesis, the solid supports were incubated at 55 °C for 6 h with NH_3_ solution (33%). The residues were analyzed by HPLC and MALDI-TOF mass spectrometry (see Appendix A). Obtained molecular weight values and retention times are shown in Appendix A.

### 4.2. Native Polyacrylamide Gel Electrophoresis (PAGE)

Native PAGE experiments were carried out in a non-denaturing 12% acrylamide gel prepared with 50 mM HEPES buffer (pH 6), 10 mM MgCl_2_ and 5% glycerol. Oligonucleotide samples were dissolved in 25 mM sodium phosphate buffer (pH 6) to a final strand concentration of 100 µM in 20 µL. Samples were annealed by heating to 85 °C followed by slow cooling to room temperature over several hours. Prior to loading, samples were supplemented with 12% glycerol. Electrophoresis was performed at a constant voltage of 140 V for 3 h using a running buffer containing 50 mM HEPES (pH 6) and 10 mM MgCl_2_. Finally, the gel was stained with a 0.005% Stains-All solution for 30 min and then visualized using an Amersham Imager 680 (GE Life Sciences, Marlborough, MA, USA). Appendix A shows the presence of a higher mobility band that is strongly positive to Stains-all in both FR1 and FR2 samples similar to control unmodified TC_5_. This band is assigned to the i-motif tetramer. On the other hand, FRfull only shows a faint band in the tetramer position.

### 4.3. UV-Monitored Studies

Absorbance versus temperature curves of i-motif structures were measured at 20 μM strand concentrations in 25 mM sodium phosphate (NaPi) buffer at pH 5 and 6. Experiments were performed in Teflon-stoppered 1 cm path-length quartz cells with a JASCO V-650 spectrophotometer connected to a Peltier accessory. The samples were heated to 85 °C, allowed to cool slowly to 25 °C and kept on the fridge overnight. Denaturation experiments were performed at 1.0 °C min^−1^ rate between 15 and 80 °C for the pH 5 experiments and from 10 to 80° for the pH 6 experiments, monitoring the absorbance at 265 nm. The data were analyzed, and the dissociation temperatures were determined as the midpoint of the transition (T_½_).

### 4.4. NMR Experiments

Samples for NMR experiments were suspended in 250 µL of H_2_O/D_2_O 9:1 in 25 mM sodium phosphate buffer. NMR spectra were acquired on Bruker Neo spectrometers operating at 600, or 800 MHz, equipped with cryoprobes, and processed with Topspin v4.1.1 software. TOCSY spectra were recorded with standard MLEV17 spinlock sequence and with 80 ms mixing time. NOESY spectra in H_2_O were acquired with 150 and 250 ms mixing times and 2048 and 512 point in t2 and t1 dimensions, respectively. NOESY experiments were acquired with and without ^19^F decoupling since combinations of the two experiments allow a rapid identification of 2′F-riboC ^1^H resonances. ^19^F detected HOESY spectra were used to assign ^19^F resonances and obtain heteronuclear fluorine-proton distance constraints. HOESY spectra were recorded with mixing times of 200 ms, and 2048 and 128 data points in t2 and t1 dimensions, respectively. Two-dimensional experiments were carried out at 5 °C. The spectral analysis program Sparky [54] was used for semiautomatic assignment of the NOESY cross-peaks.

### 4.5. Experimental NMR Constraints

Distance constraints were obtained from a qualitative estimation of NOE intensities. NOEs were classified as strong, medium or weak, and distance constraints were set accordingly to 3, 4 or 5 Å. In addition to these experimentally derived constraints, hydrogen bond constrains for the C:C^+^ base pairs were used. Target values for distances and angles related to hydrogen bonds were set to values obtained from crystallographic data in related structures [23]. Due to the relatively broad line-widths of the sugar proton signals, J-coupling constants were not accurately measured, but only roughly estimated from DQF-COSY cross-peaks. Absence or very weak H1′-H2′ cross-peaks in ^19^ F decoupled TOCSY experiments was considered as an indication of North sugar conformation. In these cases (all except T1), the sugar dihedral angles δ, ν_1_ and ν_2_ were constrained to the North domain. No backbone angle constraints were employed. Distance and dihedral angle constraints with their corresponding error bounds were incorporated into the AMBER potential energy by defining a flat-well potential term. Force constants were set to 20 kcal/mol·Å^2^ for experimental distance constraints, and to 30 kcal/mol·Å^2^ for angular constraints.

### 4.6. Structure Determination and Analysis

Structures were calculated with the SANDER module of the molecular dynamics package AMBER [55]. The coordinates of related tetrameric i-motif with the corresponding modifications to include 2′F-riboC residues were taken as starting points for the AMBER refinement, consisting of an annealing protocol in water, followed by trajectories of 5 ns each in which explicit solvent molecules were included and using the Particle Mesh Ewald [56] method to evaluate long-range electrostatic interactions. The BSC1 force field [57] was used to describe the DNA and the TIP3P model was used to simulate water molecules [58]. Force-field parameters for 2′F-araC sugars were obtained as described previously [41]. Analysis of the representative structures as well as the MD trajectories was carried out with the programs Curves V5.1 [59], X3DNA [60], VMD [61] and MOLMOL [62].

## 5. Conclusions

In summary, 2′F-riboC substitutions do not stabilize i-motif structures, yet they permit high-resolution structure determination under acidic conditions. The exclusive formation of 3′E topologies in iFRNA variants and the preservation of characteristic i-motif features support the conclusion that 2′F is less detrimental than 2′-OH. However, it lacks the unique stabilizing advantages of 2′F-araC. These findings refine our understanding of sugar-modified i-motifs and underline the subtle effects of 2′-position substitutions in tuning i-motif behavior.

## Figures and Tables

**Figure 1 molecules-30-03561-f001:**
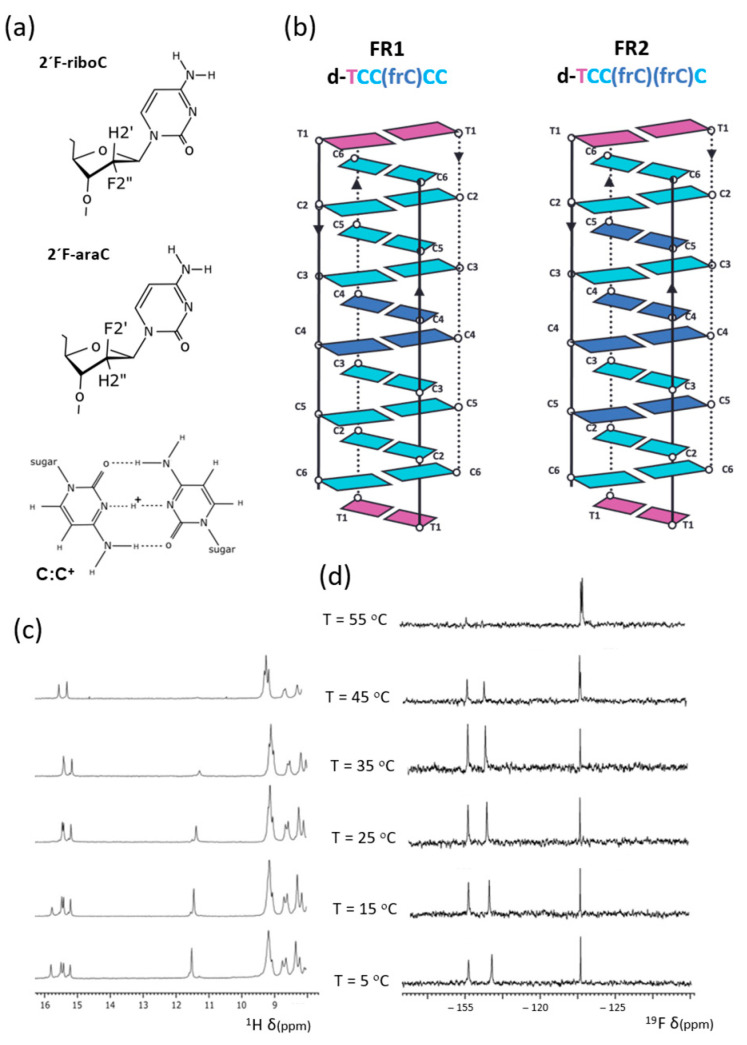
(**a**) 2′F-riboC, 2′F-araC and C:C+ base pair, (**b**) schemes of the tetrameric structures of FR1 and FR2; (**c**) ^1^H-NMR (exchangeable protons region) and (**d**) ^19^F-NMR spectra of FR2 at different temperatures (H_2_O/D_2_O 9:1, 25 mM sodium phosphate, pH 5).

**Figure 2 molecules-30-03561-f002:**
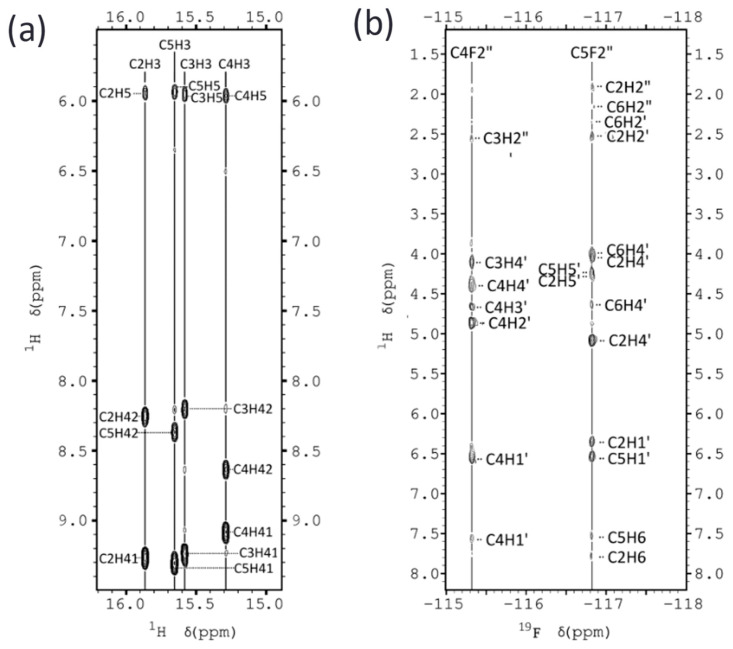
(**a**) Exchangeable protons region of the NOESY spectra of FR2, showing imino-amino cross-peaks. (**b**) ^19^F-^1^H HOESY spectra of FR2, indicating the most relevant ^19^F-^1^H cross-peaks. (H_2_O/D_2_O 9:1 25 mM sodium phosphate, pH 5, T = 5 °C).

**Figure 3 molecules-30-03561-f003:**
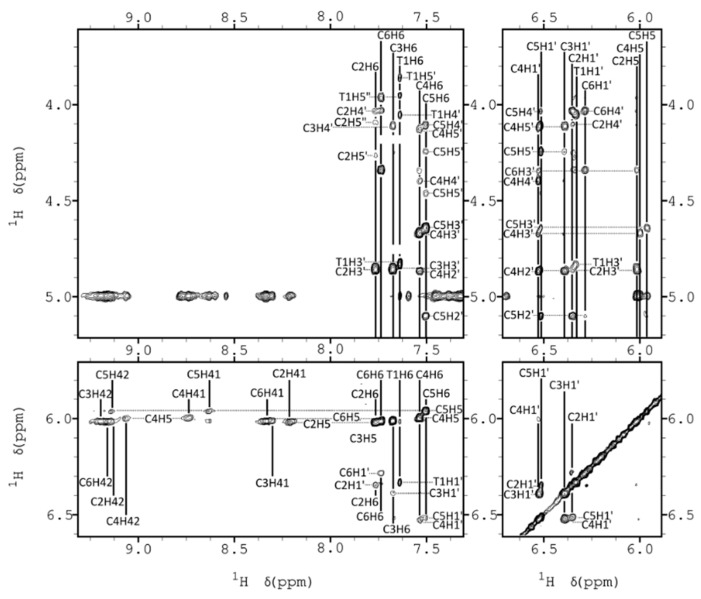
Several regions of 2D NOESY spectra of FR2, showing key NOEs (H_2_O/D_2_O 9:1 25 mM sodium phosphate, pH 5, T = 5 °C).

**Figure 4 molecules-30-03561-f004:**
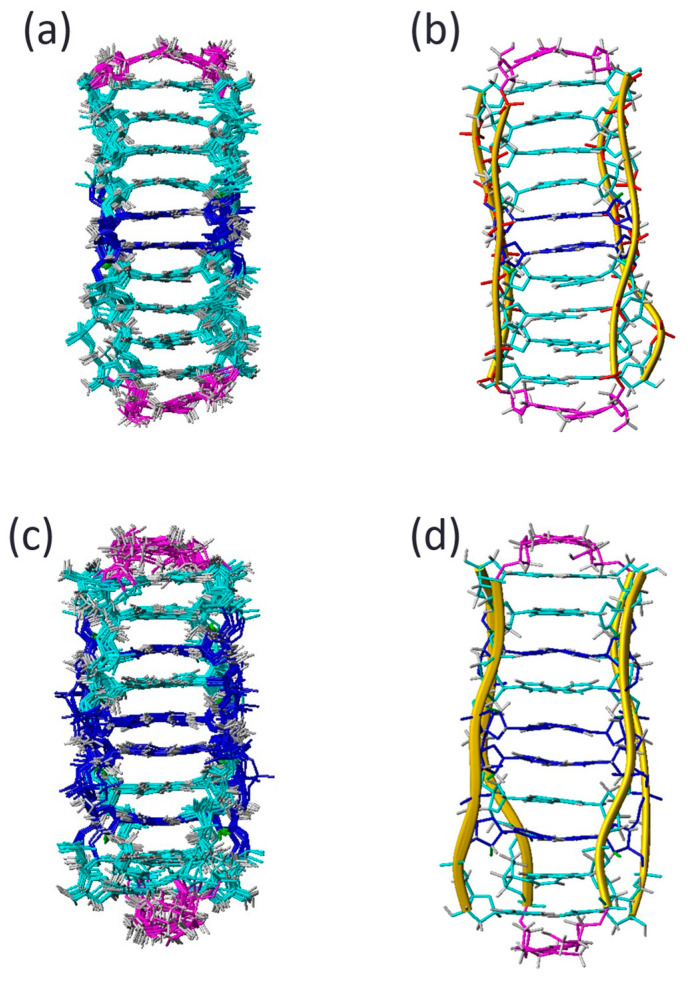
Ensemble of ten resulting structures of FR1 (**a**), FR2 (**c**) and average structures (**b**,**d**), respectively.

**Figure 5 molecules-30-03561-f005:**
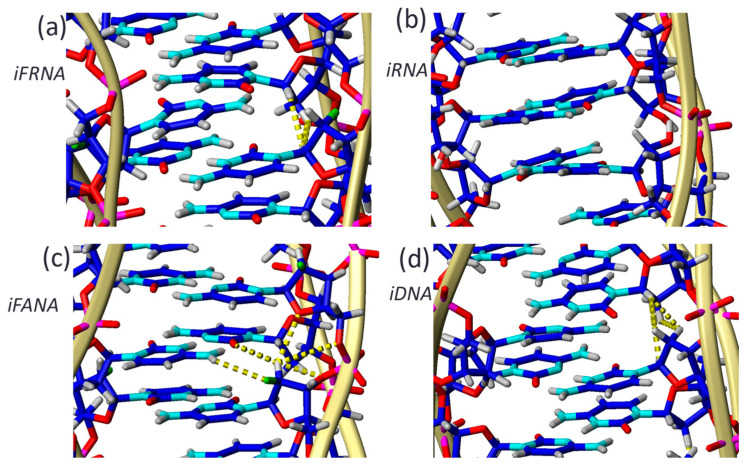
Structural details of i-motifs containing (**a**) 2′F-riboC (iFRNA), (**b**) riboC (iRNA), (**c**) 2′F-araC (iFANA), and (**d**) deoxyC (iDNA) residues. All sugars adopt a North (C3′-endo) conformation except for 2′F-araC, which assumes a South (C2′-endo) pucker. This conformation promotes favorable electrostatic interactions between the fluorine atom and polarized hydrogen atoms (indicated by yellow dashed lines). In contrast, stabilization in the iDNA structure (**d**) primarily arises from hydrophobic sugar–sugar interactions (mina contacts are shown also in yellow lines). These interactions are partially disrupted in the iFRNA structure (**a**) and largely abolished in the iRNA variant (**b**).

**Figure 6 molecules-30-03561-f006:**
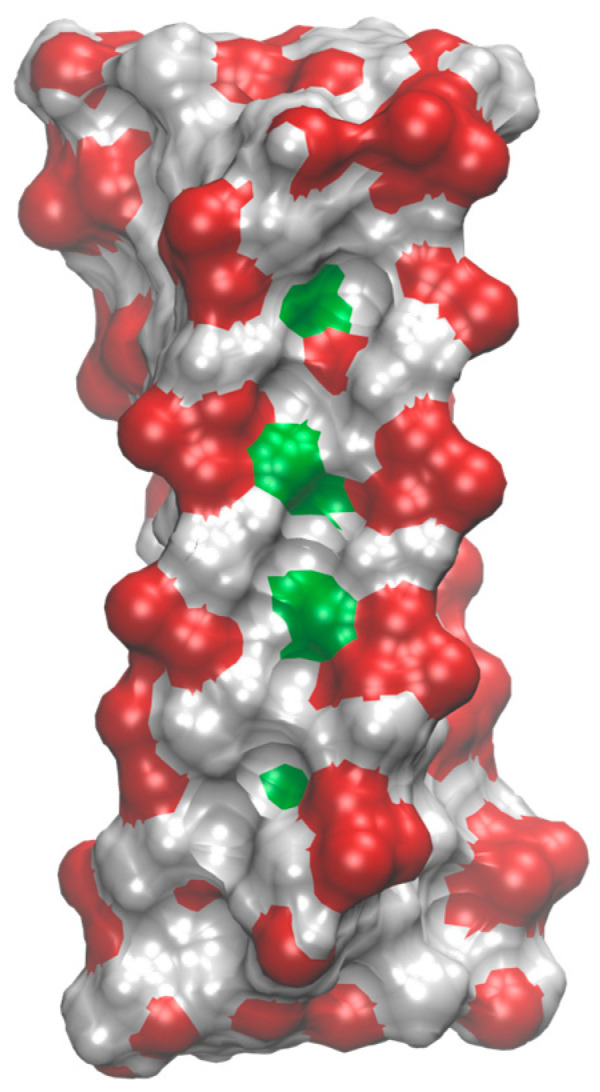
Molecular surface of FR2 i-motif structure colored according to atom types (phosphates in red, fluorine atoms in green, all other in grey). Fluorine atoms are partially exposed in the interior of the minor groove.

**Table 1 molecules-30-03561-t001:** T_1/2_ values for the FRNA modified TC5 sequences ^1^.

Name	Sequence (5′-3′)	T_1/2_ pH 5 (°C)	T_1/2_ pH 6 (°C) ^2^
FR1	dTCC(frC)CC	55.9	-
FR2	dTCC(frC)(frC)C	64.8	-
FRfull	dT(frC)(frC)(frC)(frC)(frC)	-	-

frC = 2′F-riboC. ^1^ Oligonucleotide concentration was 20 µM in 25 mM NaPi buffer at pH 5 and 6. ^2^ Transitions with small hyperchromicity were observed for FR1 and FR2 sequences at low temperatures.

## Data Availability

The data underlying this article are available in the Protein Data Bank (PDB) at https://www.rcsb.org/, and can be accessed with IDs 9S6Z and 9S6I for FR1 and FR2, respectively (Extended PDB ID pdb_00009S6Z, BMRB ID 35011 and PDB ID 9S6I, Extended PDB ID pdb_00009S6I, BMRB ID 35010). All other data are available in the article or Appendix A. Raw data will be shared on request to the corresponding/first authors.

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
