# Peer review of "The Effect of 2′F-RNA on I-Motif Structure and Stability"

_molecules, 2025, doi:10.3390/molecules30173561_

Round 1

Reviewer 1 Report

Comments and Suggestions for Authors

Ugedo et al. reported here their investigations of the effects of incorporating 2′-fluoro-ribocytidine (2′F-riboC) into i-motif-forming DNA sequences. It was revealed that full substitution destabilizes the structure, while partial substitutions (FR1 and FR2) maintain stable i-motif formation at acidic pH. Further structural analysis via NMR reveals conserved C•C⁺ base pairing and a North sugar pucker orientation, positioning the fluorine substituent in the minor groove. The findings highlight the utility of 19F NMR for studying nucleic acid dynamics and demonstrate how fluorine modifications can modulate i-motif stability.

This study presents findings of broad interest, with well-supported conclusions and a logically structured presentation. However, several key issues should be addressed to strengthen the manuscript for publication.

Specific Recommendations:

  1. Abstract Clarity: The full names of abbreviations (e.g., NOEs) should be provided upon first use.
  2. Formatting Consistency: Figures, tables, and scheme labels should be bolded for better readability.
  3. Figure 1 Reorganization: The current layout could be improved for clarity—consider adjusting panel arrangement.
  4. Oligonucleotide Characterization: Additional data (e.g., HPLC, MS) confirming the purity and identity of FR1/FR2/FRfull oligos should be included.
  5. NMR Experimental Details: The spectrometer frequency (MHz), deuterated solvent, and acquisition conditions (e.g., temperature, buffer) must be specified for reproducibility.
  6. Structural Accuracy in Figure 1A: re-draw the chemical structure.
  7. Crystallography Methods: Detailed conditions (e.g., crystallization setup, resolution, refinement statistics) are missing and should be provided.
  8. Supplementary Materials: Essential supporting data (e.g., additional spectra, gel electrophoresis) should be included as supplementary information.

Author Response

Thank you very much to both referees in reviewing our manuscript. We sincerely believe that their observations have significantly contributed to improving the quality of our study.
Please find below a summary of the changes made to the manuscript according to both referees's suggestions:

- Abbreviations have been removed from the abstract.
- Figure 1 has been revised, and the scheme in Figure 1a has been modified for clarity.
- Additional data have been included in the Supplementary Material, such as native gel electrophoresis and mass spectrometry characterization. The corresponding figures (4) have been added, together with a new paragraph in the Materials and Methods section.
- More details on the NMR experiments have been provided, including the deposition of the coordinates of the determined structures in the PDB (codes 9S6Z and 9S6I) and all relevant NMR data in the corresponding BMRB database. All of these data will be available upon publication.
- The discussion on the potential role of the 2′F-riboC modifications in i-motifs has been expanded, with further explanation of the possible reasons for their increased stability compared to i-motif RNAs, as well as the potential implications of including fluorine derivatives for i-motif detection in vivo.
- Additional references have been included, and several self-citations from our laboratories have been removed. References to our previous work now represent less than 20% of the total. In any case, it should be noted that this study is the result of a collaboration between two very active groups in this field, which makes some degree of self-citation difficult to avoid.

Reviewer 2 Report

Comments and Suggestions for Authors

Ugedo et al., in the present study explored the effect of fluorine substitution on i-motif structure and stability and their impact on 19F NMR spectroscopy. The authors provided with high-resolution three-dimensional structures of two i-motif variants, shedding light into the molecular basis of the destabilizing effect induced by fluorine in the ribo-configuration and their value in 19F NMR.

The article is interesting and well written and may have clinical implications.

It would be helpful if the authors could also discuss how hybrid fluorine-modified iMs formed with DNA and/or RNA could be detected and what are the biological implications of RNA  iMs in vivo.

The authors should include gel electrophoresis analysis to provide with information about oligonucleotide concentration and annealing.

Author Response

(The authors gave the same response as above.)
